# A Novel Method for Controlled Gene Expression via Combined Bleomycin and Plasmid DNA Electrotransfer

**DOI:** 10.3390/ijms20164047

**Published:** 2019-08-19

**Authors:** Sonam Chopra, Paulius Ruzgys, Milda Jakutaviciute, Aiste Rimgailaite, Diana Navickaitė, Saulius Satkauskas

**Affiliations:** Biophysical Research Group, Faculty of Natural Sciences, Vytautas Magnus University, Vileikos st. 8, LT-44404 Kaunas, Lithuania

**Keywords:** gene electrotransfer, cell death, bleomycin, plasmid DNA, transfection, cytotoxic effect

## Abstract

Electrochemotherapy is an efficient method for the local treatment of cutaneous and subcutaneous metastases, but its efficacy as a systemic treatment remains low. The application of gene electrotransfer (GET) to transfer DNA coding for immune system modulating molecules could allow for a systemic effect, but its applications are limited because of possible side effects, e.g., immune system overactivation and autoimmune response. In this paper, we present the simultaneous electrotransfer of bleomycin and plasmid DNA as a method to increase the systemic effect of bleomycin-based electrochemotherapy. With appropriately selected concentrations of bleomycin and plasmid DNA, it is possible to achieve efficient cell transfection while killing cells via the cytotoxic effect of bleomycin at later time points. We also show the dynamics of both cell electrotransfection and cell death after the simultaneous electrotransfer of bleomycin and plasmid DNA. Therefore, this method could have applications in achieving the transient, cell death-controlled expression of immune system activating genes while retaining efficient bleomycin mediated cell killing.

## 1. Introduction

Electroporation is a widely used technique that increases the permeability of the cellular membrane by exposing the cells to electric fields of appropriate intensity and duration [1,2]. This technique has been positively accepted by scientific and medical communities due to its safety and effectiveness for transferring nucleic acids, drugs, etc., for pharmacological studies and clinical applications (in particular, oncological applications) [3,4].

Electroporation-induced changes in cell permeability can be classified as reversible and irreversible electroporation, according to the membrane resealing of affected cells [5]. In the case of reversible electroporation, the cell membrane reseals after transient permeabilization [6]. Reversible electroporation can be used in clinics to facilitate the entry of chemotherapeutic agents such as bleomycin for the local treatment of tumors (this process is known as electrochemotherapy (ECT)) [7]. ECT is one of the prominent treatments in oncology and is used in several oncology centers throughout Europe for the palliative treatment of skin and subcutaneous metastases of many cancer types [8,9,10]. ECT effectiveness in antitumor treatment depends on many factors, including the tumor type [11,12] and the anticancer drug used [13]. As one of the most commonly used electro chemotherapeutic drugs, bleomycin acts as a cleaving agent for cellular DNA molecules [14], and its action also results in the induction of endoplasmic stress. Consequently, this may lead to the immunogenic cell death and the activation of the anti-cancer immune response [15].

It is known that immune response activation helps in the complete eradication of tumor cells [16]. However, conventional ECT does not induce enough stimulation of the immune system to prevent the formation of new tumors or the growth of distant tumors. In order to boost the immune response of electrochemotherapy, a stimulating agent is needed [17,18]. For this reason, it has been proposed to use gene electrotransfer in combination with ECT for the enhancement of the systemic antitumor effect [19,20]. It is already been proven that combining ECT with immunostimulating agents increases the response rate of distant tumors in deep-seated regions—not only the tumors treated by ECT [3]. Though there are many scientific publications claiming the effective treatment of cancer by inducing the anti-tumor immune response [4,21,22,23], there are still many challenges to overcome in order to achieve safe and efficient tumor immunotherapy. While the efficiency of cancer immunotherapies still leaves something to be desired [24,25], the main barrier to the application of immunotherapy is the possibility of immune-related adverse effects (for reviews see [26,27]). Some authors claim that cancer immunotherapy employing the delivery of the cytokine-coding vector has less side effects than the direct delivery of the same cytokine [28,29]. However, it should be noted that its transfection efficiency and its related treatment efficiency is still low, and current trials aim to improve it [30]. Therefore, with the increasing efficiency of transfection, concerns about the over-expression of therapeutic cytokines after immunogenetherapy and associated adverse reactions remain [31]. This shows the need to develop techniques that allow for the precise control of gene expression after the delivery of genes coding immune system stimulating molecules.

There are already some techniques that allow for the suppression of protein production, e.g., RNA interference using small interfering RNAs (siRNAs) [32,33] or micro RNA (miRNA) mimics [34,35]. However, these techniques also suffer from a plethora of issues, including limited transfection efficiency, off-target effects and adverse immune reactions [36]. Additionally, applying these techniques to modulate the expression of a previously transfected gene would need a separate procedure, increasing the complexity, duration and cost of the treatment.

In this paper, we present a novel method that allows for cell death-regulated gene expression. This method relies on the simultaneous electrotransfer of the anticancer drug bleomycin and plasmid DNA, and it allows for control over the transfection efficiency and cell death dynamics by varying the concentration of bleomycin used. To showcase this method, we analyzed different concentrations of bleomycin and plasmid DNA in order to design an optimal combination, achieving a high level of cell death from bleomycin while maintaining localized temporary gene expression. The transfection efficiency and cell death were evaluated at different time points post-treatment. These findings may be perceived as a leading way to increase therapeutic outcomes among a range of different tumors.

## 2. Results

Firstly, a set of experiments was performed to check transfection efficiency and cell viability on the dependence of plasmid concentration. For these experiments, cells were electroporated with different concentrations of green fluorescent protein (GFP)-coding plasmid (10, 50 100, 200 and 400 µg/mL) using one high-voltage (HV) pulse. The results of the experiments are shown in Figure 1A. At a 10 µg/mL plasmid concentration, the amount of GFP-positive cells was too low. However, the increase of plasmid concentration from 100 to 400 µg/mL yielded a significantly higher number of GFP-positive cells. Though increasing the plasmid concentration to 400 µg/mL increased the number of GFP-positive cells, the cell viability in these conditions decreased, as seen in Figure 1B. Since the 200 µg/mL plasmid concentration was enough to get a high number of GFP-positive cells while maintaining cell viability, we used a 200 µg/mL plasmid concentration in all the experiments described below.

After optimizing the plasmid DNA concentration, we tested the effect of extracellular DNA on the efficiency of bleomycin electrotransfer for a range of bleomycin concentrations (0.1–20 nM). It is known that bleomycin has very low membrane permeability and enters cells easily using the electric pulses. This results in cell death, which depends on the number of permeabilized cells and the bleomycin concentration. The results are presented in Figure 2A. There was no significant difference between cells electroporated in the presence or the absence of extracellular plasmid DNA when a 15 nM or higher concentration of bleomycin was used. However, at a bleomycin concentration of 0.5 nM, 70% of cells died in the presence of extracellular plasmid DNA, but only 55% of cells died in the absence of plasmid DNA. This shows that, at lower bleomycin concentrations, extracellular DNA has a clear effect on bleomycin electrotransfer. Afterwards, we used a comet assay to quantify the DNA damage caused by bleomycin in the cells when the cells were electroporated in the presence and the absence of pDNA. The damage evaluation was based on the length and fluorescence intensity of comet tails. Figure 2B shows the DNA damage assayed by the comet assay immediately after electroporation with bleomycin either alone or in the presence of pDNA. The results revealed an increase in DNA damage when bleomycin electrotransfer was performed in the presence of plasmid DNA. When a bleomycin concentration of 10 nM and above was used in combination with plasmid DNA, significant DNA damage was observed. Around 10% of DNA damage was observed when electroporation was done with bleomycin in the presence of plasmid DNA, but only 5% of DNA damage was observed when electroporation was done with bleomycin only at these bleomycin concentrations.

Comet categories are defined by the fluorescence intensity of the tail as well as tail length. The extents of DNA damage at different bleomycin concentrations in the absence or presence of extracellular DNA are illustrated in Figure 3. It could be argued that the extra DNA damage visible comes from the plasmid DNA, but our calculations showed that this is unlikely. We used a formula from [37] to calculate the volume from which DNA could come to the plasma membrane of the cell during electric pulses and interact with it. Knowing the length (3486 bp) and the mass (2.15 × 10^6^ g/mol) of the plasmid, we estimated that this volume contained 4523 plasmids (or a total of 1.58 × 10^7^ bp) of plasmid DNA per cell. However, this number was realistically smaller because (a) DNA delivery to the cell is more complicated than DNA just being present in the vicinity of electroporated cell membrane, and (b) only a fraction of cell membrane would reach high enough transmembrane potential to be electroporated. Nevertheless, even using the inflated values of plasmid DNA that entered the cell, the length of genomic DNA in Chinese hamster cells (2.23 × 10^9^ bp according to the representative assembly in NCBI database) was 147 times larger than that of all the plasmid DNA entering the cell. Realistically, this number would be much higher due to the limitations of plasmid entry listed above. Therefore, a significant plasmid DNA contribution to the comet tails was very much unlikely.

Previous studies reported that the electrotransfer of bleomycin results in cell death at various times periods [38]. Due to this reason, the flow cytometer assay was employed in the study to estimate the exact number of cells in different concentrations at 12, 24, 48, and 72 h time points after the experiment. The results from these experiments are shown in Figure 4. The cell viability after bleomycin electrotransfer in the presence of plasmid DNA measured with a clonogenic assay (CA) was significantly different from cell viability measured with a flow cytometry assay at all the time points with bleomycin concentrations 0.5 nM and higher. At these aforementioned time points, cell viability was higher in comparison with the cell viability evaluated by the CA, which illustrates that at these bleomycin and pDNA concentrations, bleomycin-induced cell death is a slow process, with some of the cells dying up to and above 72 h post treatment. It can be seen that with the higher doses of bleomycin, cell death was quicker. For example, the reduction of viable cells to 30% took up to 24 h with 5 nM and higher bleomycin concentrations, the reduction took up to 48 h with a 1 nM bleomycin concentration, and it took up to 144 h (6 days) with a 0.5 nM bleomycin concentration. With 5 nM and higher bleomycin concentrations, ~50% of the cells died within first 12 h of the treatment. Meanwhile, after treatment with 0.5 nM bleomycin, no cell death was observed 12 h after treatment, and the reduction of surviving cells to ~50% took up to 48 h. Additionally, around 30% of the cells died between 72 h and 144 h (six days) after treatment with 0.5 nM bleomycin and pDNA.

Later, we analyzed transfection efficiency after bleomycin and pDNA co-electrotransfer. However, since a major decrease in cell viability was observed in some of the points, we included the cell viability in the transfection efficiency calculations, yielding the percentage of GFP-positive cells (Figure 5A) and total fluorescence (Figure 5B) as calculated from all treated cells. In all the samples, both the percentage of transfected cells and total fluorescence reached its peak 24 h after the treatment, which is the standard time for evaluation of transfection efficiency. In the control sample (pDNA with 0 nM bleomycin), the percentage of GFP-positive cells remained constant 24–72 h post treatment. However, in all of the bleomycin and pDNA groups, the percentage of GFP-positive cells decayed over time. As it was already seen that bleomycin, at concentrations used, caused cell death (Figure 4), we can conclude that the decrease of GFP-positive cells was due to bleomycin-induced cell death, showing a possibility to achieve the temporal control of DNA expression by bleomycin.

## 3. Discussion

In the results described above, we show the effects of the simultaneous electrotransfer of bleomycin and plasmid DNA. Firstly, we show that the addition of plasmid DNA increased the efficiency of bleomycin electrotransfer to the cells, both in the number of cells killed and in the extent of DNA damage present in the cells (see Figure 2 and Figure 3). We already showed the increased efficiency of small molecule electrotransfer when DNA is present in the electroporation medium in our recent paper [39], but the results in Figure 2 and Figure 3 also show that the DNA increased not only the number of cells affected but also the degree of DNA damage in the affected cells. This serves as a further proof that more bleomycin molecules enter the cells when DNA is present in the medium, decreasing the concentration of bleomycin needed to achieve the same cytotoxic effect. One of the possible reasons is related to the nature of bleomycin. As the bleomycin molecule exerts its cytotoxic effect on the cell by binding the DNA and facilitating single and double-strand cleavage [14,40], it is expected that the bleomycin molecules bind the extracellular DNA when both DNA and bleomycin are present in the extracellular medium before electroporation. Therefore, a fraction of bleomycin might enter the cell together with the DNA, separately from the bleomycin that enters the cell via diffusion. Because the plasmid DNA needs to reach the nucleus of the cell for the transfection to happen [41,42], it could also facilitate the nuclear entry of this fraction of bleomycin molecules. Because single bleomycin molecule can cleave the DNA ~10 times [43,44,45], the bleomycin molecules that enter the cell while bound to plasmid DNA could still be re-activated and cleave the DNA inside the cell. Alternatively, it is possible that the presence of DNA-membrane complexes changes the local properties around the pore area, facilitating the small molecule transfer into the cells.

In the light of the DNA-cleaving nature of bleomycin, it is tempting to think that the DNA would be used solely as the adjuvant to help the entry of bleomycin into the cells. However, our results show that this is surprisingly not the case, as evident in Figure 5. Though it is true that the cells eventually die to the cytotoxic effect of bleomycin when the dose of bleomycin is sufficient, as seen in Figure 2 and Figure 4, this cell killing effect is not instantaneous. It has been shown by [46] that bleomycin kills the cells in two different ways, dependent on the intracellular bleomycin concentration: A high amount of intracellular bleomycin causes the cell to die apoptotically, and a low amount of intracellular bleomycin causes slower mitotic cell death. However, neither apoptosis nor mitotic cell death are instantaneous processes, which means that the cells will not immediately die after bleomycin enters the cell. The dynamics of bleomycin-induced cell death are seen in Figure 4, and it is evident that the onset of the cell death for the majority of the cells depends on the bleomycin concentration used. At a 0.5 nM concentration of bleomycin, only a small amount of cells (~10%) died in the first 12 h post bleomycin electrotransfer; instead, the bulk of the cells that died did so slowly, as seen by the gradual decrease of cell number in the three days post bleomycin electrotransfer. In this case, around 40% of cells died within 72 h of bleomycin electrotransfer, and another 20% died between three and seven days post bleomycin electrotransfer. In contrast, at a 20 nM bleomycin concentration, ~70% of cells died within twelve hours post bleomycin electrotransfer, and after 48 h, there were barely any cells left. This shows that the concentration of bleomycin allows for control over the dynamics of cell death. Thus, the cells can survive long enough for the expression of proteins encoded in the plasmid DNA to take place.

It is apparent from Figure 5 that not all the plasmid DNA is degraded before its entry into the cell nucleus. One of the possible reasons for this is the mode of DNA-membrane complex formation and bleomycin binding to DNA. As bleomycin binds the DNA in a way that prevents nuclease binding, this suggests competitive binding of DNA nucleases and bleomycin [47]. It has also been shown that DNA becomes inaccessible to the effect of nucleases [48] and intercalating dyes [49] after one min post the formation of DNA-membrane complexes. Thus, the formation of DNA and membrane complexes would likely prevent further binding of bleomycin to the plasmid DNA and, thus, the dynamics of the bleomycin-induced DNA degradation. Additionally, with the concentrations used in our study, it was evident that there was a surplus of pDNA in comparison to bleomycin molecules. We calculated the ratio of pDNA and bleomycin molecules when pMAX-GFP plasmid was used at a 200 µg/mL concentration and bleomycin was used in 0.1–20 nM. At the 50 µL sample, there were 2.79 × 10^12^ plasmid molecules and 6.02 × 10^8^ bleomycin molecules at a 20 nM bleomycin concentration, which led to a 1/4640 bleomycin/DNA molecule ratio. At the 0.1 nM bleomycin concentration, this ratio decreased to 1/928,184. Even if the action of bleomycin molecules is swift and efficient, it is more than likely that some of the plasmid DNA molecules manage to stay intact during the process of electrotransfection.

As we have shown that both the controlled timing of cell death post bleomycin treatment and DNA electrotransfection is possible, this paves the way for the cell-death controlled transfection of cells. The application of combined bleomycin-DNA treatment would allow for the transient expression of the desired gene in the treated cells, which would be later “silenced” because the affected cells would die. This could prove useful in combined immunochemotherapy, which has been a rising field in cancer treatment as of late [15]. Gene therapy using plasmids coding for immune system molecules has shown promising results in clinical trials as well as in animal models [50,51,52]. One of the main cytokines that has been investigated for potential in cancer immunotherapy is interleukin-12 (IL-12) [53,54,55]. Interferon-α [56] and interleukin-2 [57,58] have also been investigated as candidate genes for cancer immune-gene-therapy. Many studies have been trying to increase the transfection efficiency of immune system molecule-coding vectors [59,60]. However, it should be noted that the non-controlled high transfection efficiency of these molecules does not necessarily convey a good therapeutic effect [61]. Firstly, an elevated concentration of immune system molecules might lead to deleterious adverse reactions, similar to those observed after the systemic administration of cytokines [62,63]. In addition, the prolonged expression of immune system molecules might not necessarily equate to a better immune system response. There are experimental data showing that long (over 48 h) T cell stimulation does not increase the T cell activity, but it does decrease the viability of those cells [64]. However, there are a regrettable lack of data concerning the onset of immune response after cytokine gene therapy, so future in vivo studies investigating this gap in knowledge might be needed. As the method presented in this paper shows the possibility of the temporal regulation of transfection efficiency at different time points post-transfection, dependently on bleomycin concentration, it could be possible to employ our method for these types of studies.

The simultaneous application of both chemotherapeutic drugs and the cytokine-coding plasmid holds one additional improvement over electrochemotherapy followed by electrogenetherapy that was proposed several years ago [49,65]. We believe that the combined simultaneous treatment is advantageous not only because of its ability for time-resolved transfection, but also because it could reduce the amount of potentially invasive treatments needed. This is especially attractive for the treatment of deep-seated tumors. Therefore, the novel method proposed here could help to reduce the time and cost needed for the treatment, increasing the applicability and the comfort of the patient.

## 4. Materials and Methods

### 4.1. Cell Culture

Chinese hamster ovary (CHO) cells (European Collection of Authenticated cell cultures, 85050302) were used for the experiments. Cells were grown in Dulbecco’s Modified Eagle Medium (DMEM), supplemented with 1% penicillin-streptomycin, 1% l-glutamine and 10% FBS. All reagents were purchased from Sigma-Aldrich (Darmnstadt, Germany). The cells were grown in a 96 mm culture dish at 37 °C in a humidified incubator at 5% CO_2_. To maintain the culture, cells were passaged every 2–3 days and always a day before an experiment.

### 4.2. Plasmid DNA Preparation

The pMAX-GFP plasmid (3486bp) (Amaxa, Cologne, Germany) coding for enhanced green fluorescent protein was used in the experiments. Plasmid concentrations of 0–400 µg/mL were used to enhance the transfer of the chemotherapeutic drug, bleomycin, and to evaluate the percentage of transfected cells. The plasmid was purified from competent DH5α cells using endo-free plasmid Giga Prep kit (Qiagen, Valencia, CA, USA), according to the manufacturer’s instructions. The concentration and the quality of the plasmid was checked using a spectrophotometer (Nanodrop 2000, Thermo Fisher, Washington, DC, USA) and gel electrophoresis.

### 4.3. Cell Electroporation

For electroporation, the cells were trypsinized and resuspended in a laboratory-made electroporation medium that had 270 mOsm osmolarity, 0.1 S/m conductivity, and 7.1 pH. The electroporation medium was made from sucrose (242 mM), Na_2_HPO_4_ (5.5 mM), NaH_2_PO_4_ (3 mM) and MgCl_2_ (1.73 mM). The cell concentration was set to be 2 × 10^6^ cells/mL for experiments using DNA alone and 2.25 × 10^6^ cells/mL for the experiments using the combination treatment (bleomycin (BLM) and pDNA). For experiments with the plasmid alone, 45 μL of cell suspension (9 × 10^4^ cells) was supplemented with either 5 μL plasmid DNA (to a 10, 50,100, 200 or 400 μg/mL final concentration) or 5 μL of the electroporation medium (control). For the combination experiments (BLM and pDNA), 40 μL of cell suspension (9 × 10^4^ cells) in the electroporation medium were supplemented with 5 μL of plasmid DNA at a 200 μg/mL final concentration and 5 μL of bleomycin (0.1–20 nM) or the electroporation medium. Cells were then placed between stainless steel electrodes with a 2 mm gap between the electrodes and subjected to a single high-voltage square pulse (1400 V/cm, 100 μs); they were then left to reseal for 10 min. Electroporator BTX T820 (Harvard Apparatus, Holliston, MA, USA) was used to deliver the electric pulses.

### 4.4. The Measurement of DNA Electrotransfer Efficiency

Cells were electroporated in the presence of plasmid at different concentrations as described above. After 10 min of incubation, 950 μL of a growth medium (DMEM) was added to 50 μL of an electroporated cell suspension. Afterwards, 900 μL of a diluted cell suspension was transferred into 24 well plate (Plastibrand; Wertheim, Germany). Cells were grown for 24 h, and then the growth medium was removed. Each well was supplemented with 150 μL of 1× TryplE Express (Thermo Fisher, Dublin, Ireland) for the determination of transfection efficiency using a flow cytometer (BD Accuri C6, BD Biosciences, Franklin Lakes, NJ, USA). A 488 nm laser was used for cell excitation, and the fluorescence was collected using a 533/30 nm bandpass filter. The results were analyzed with BD Accuri C6 Software. This way the transfection efficiency (TE), evaluated as the percentage of GFP-positive cells, and the mean cell fluorescence (MCF) of the transfected cell population was obtained. The total fluorescence (TF) was calculated as shown in Equation (1):(1)$$TF=\frac{TE*10000*MCF}{100}$$

### 4.5. DNA Damage Evaluation

A comet assay was used to evaluate DNA damage. For the comet assay, the cells were transferred to 1.5 mL tubes and supplemented with 950 µL of the DMEM medium for 10 min, then incubated for 1 h at 37 °C. After this, the cells were centrifuged at 0.2 relative centrifugal force (RCF) for 3 min, supernatant was removed, and the cells were resuspended in 130 µl of 0.5% low-melting agarose dissolved in PBS. A single drop of this solution from each experimental point was placed on microscope slides pre-coated with 1% of agarose dissolved in dH_2_O and covered with glass coverslips. After this, the samples were incubated for 10 min at 4 °C, after which the coverslips were removed. The slides with the cells were then lysed by placing the slides in a Coplin jar containing 2.5 M NaCl, 100 mM EDTA, and 10 mM Trizma base (pH 10) at 4 °C for at least 12 h. After that, slides were immersed in an electrophoresis solution (10 N NaOH and 200 mM EDTA, pH >13) for 20 min. Electrophoresis was then carried out at 0.72 V/cm (10 V, 300 mA) for 30 min in the same solution. Slides were washed one time in with neutralization buffer for 5 min. Comets were fixed by immersing the slides in 98% ethanol for dehydration. The drying was finished in an oven at 500 °C for 30 min, and the slides were stored in a dry place before the image analysis. On the day of analysis, gels were hydrated by incubating the slides in distilled water for 30 min. Comets were stained with 1% 10 mg/mL ethidium bromide and then covered with a glass coverslip (24 × 60 mm). CHO cell photographs were captured with a high-resolution digital camera using Motic Images Advanced 3.2. software. The image analysis computer system CASP 1.2.3 b2 was used to evaluate the DNA breaks in the cells. The software measured the percentage of DNA in the comet’s head (intact DNA) and in the comet’s tail (small pieces of cleaved DNA). The percentage of DNA in the comet’s tail was the parameter used to describe the DNA damage in each comet. The number of comets per gel (including so-called hedgehogs) was counted by eye.

### 4.6. Evaluation of Cell Viability

A clonogenic assay (CA) was performed to evaluate end-point cell viability. In order to perform the CA, 400 cells from each experimental point were seeded to a 40 mm diameter Petri dish containing 2 mL of a growth medium for a duration of 6 days. The cells were then fixed with 70% ethanol for 10 min and then stained with crystal violet dye. The colonies were scanned with a scanner (CanoScan LiDE220, Canon, New York, USA) and counted using the CellCounter function in the open source software ImageJ (National Institutes of Health, Bethesda, MD, USA).

A flow cytometry assay (FCA) was used to determine the cell viability at short-to-medium time points after electroporation (12–72 h). The use of flow cytometry allowed for the determination of the number of cells with no ‘marker’ molecules used. 1 × 10^4^ of cells were plated in a 24-well microplate (Plastibrand; Wertheim, Germany). At the defined time points, the cells were trypsinized with 1X TrypLE Express. Using a flow cytometer, an absolute number of cells as counted in each cell suspension sample, and the number of cells in the experimental groups was expressed as the percentage of the cells present in time-matched control.

### 4.7. Statistics

Each experimental point was obtained from at least three independent experiments, and results are represented as the mean ± SEM. The statistical analysis was done using Microsoft Excel, and SigmaPlot software was used to produce the graphs. The statistical significance of differences between the groups was evaluated using a two-tailed Student’s *t*-test for independent samples. The *p* values < 0.05 were deemed to be statistically significant.

## 5. Conclusions

In summary, the combined simultaneous treatment could have applications, not only because of its ability for time-resolved transfection but also because it could reduce the amount of potentially invasive treatments needed, especially for deep-seated tumors, thus reducing the time and cost needed for procedures.

## Figures and Tables

**Figure 1 ijms-20-04047-f001:**
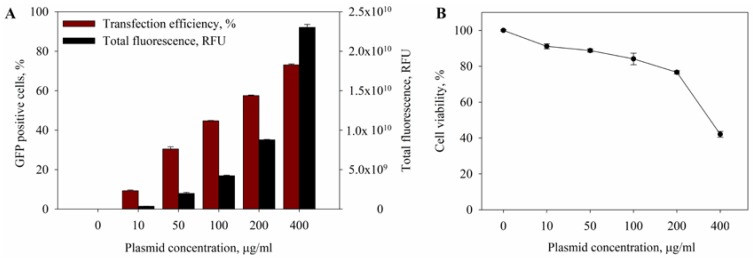
The dependence of the percentage of green fluorescent protein (GFP)-positive cells (**A**) and cell viability (**B**) on the concentration of GFP-coding plasmid. One high voltage (HV) pulse (1400 V/cm, 100 µs) was used. Error bars represent standard error of mean (SEM).

**Figure 2 ijms-20-04047-f002:**
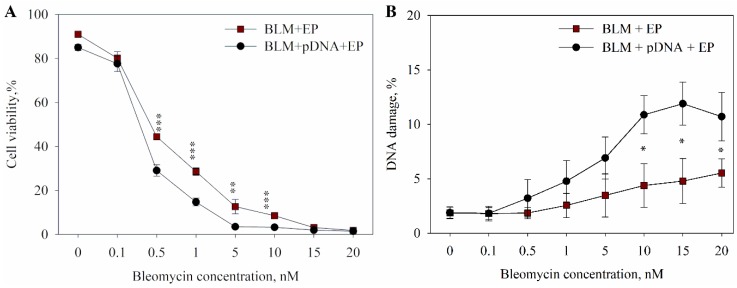
The dependence of cell viability (**A**) and DNA damage (**B**) on bleomycin (BLM) concentration and electroporation with or without extracellular plasmid DNA. The cells were electroporated with one high-voltage (1400 V/cm, 100 μs) electric pulse. The points on the curves at the 0 nM bleomycin concentration represent the control points (electroporation (EP) alone and EP with plasmid DNA). Two-tailed Student’s *t*-test *p* values < 0.05 between cells electroporated with or without extracellular plasmid DNA are marked with *, *p* < 0.01 is marked with **, and *p* < 0.001 is marked with ***.

**Figure 3 ijms-20-04047-f003:**
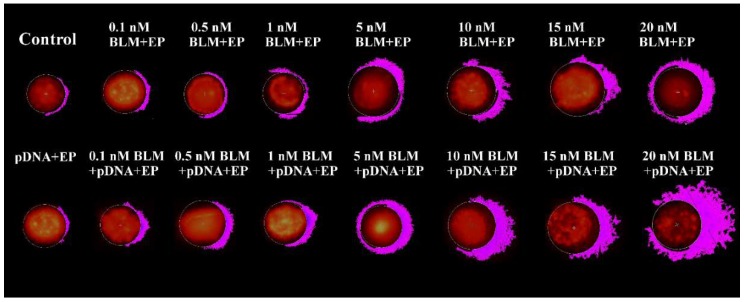
The visualization of Chinese hamster ovary (CHO) cell comets after bleomycin electrotransfer in the absence or the presence of extracellular plasmid DNA.

**Figure 4 ijms-20-04047-f004:**
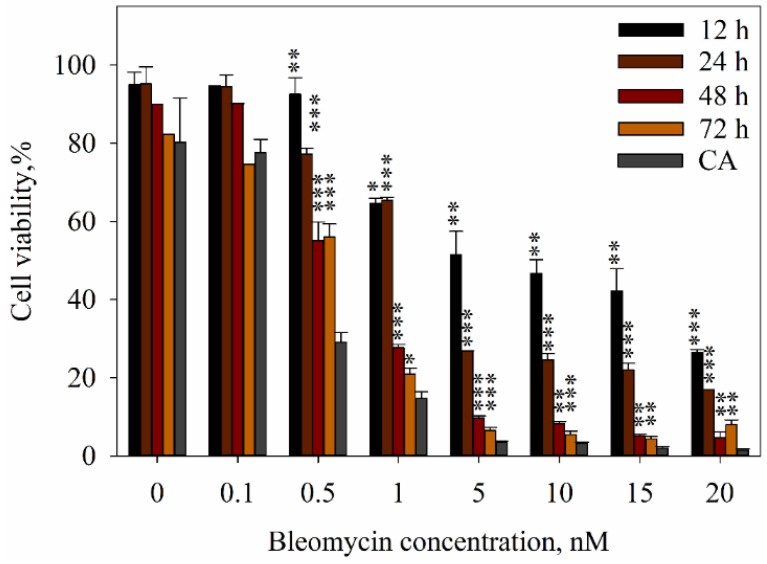
Dependence of cell viability after treatment with different bleomycin concentrations following electric pulses in the presence of plasmid DNA as assessed by a flow cytometry assay (FCA) at various time points after the treatment (12, 24, 48, 72 h) and a clonogenic assay (CA) (six days after treatment). The cells were electroporated with one high-voltage (1400 V/cm, 100 μs) electric pulse. A two-tailed Student’s *t*-test value *p* < 0.05 between the cell viability evaluated by a clonogenic assay and cell viability at different time points after electroporation is marked with *, *p* < 0.01 is marked with with **, and *p* < 0.001 is marked with ***.

**Figure 5 ijms-20-04047-f005:**
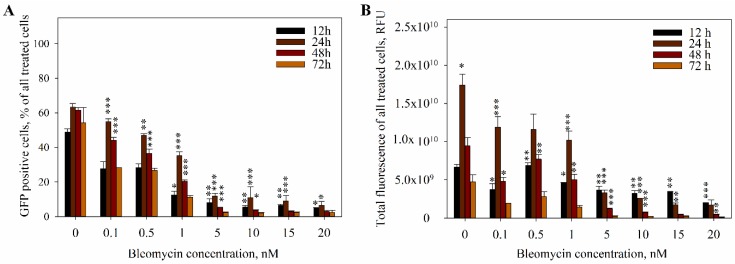
The dependence of electrotransfection efficiency (**A**) and total cell fluorescence (**B**) of all treated cells on the bleomycin concentration at various time points (12, 24, 48, and 72 h). The cells were electroporated with one high-voltage (1400 V/cm, 100 μs) electric pulse. Two-tailed Student’s *t*-test *p* values < 0.05 are marked with *, *p* < 0.01 is marked with **, and *p* < 0.001 is marked with ***.

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
