# Peer review of "A Novel Method for Controlled Gene Expression via Combined Bleomycin and Plasmid DNA Electrotransfer"

_ijms, 2019, doi:10.3390/ijms20164047_

Round 1

Reviewer 1 Report

Improvements of electrochemotheray (ECT) protocols are of pivotal importance for clinical applications. In 2006 European Standard Operating Procedures of ECT (ESOPE) were released (Mir et al., E J C  S 4 ( 2 0 0 6 ) 1 4 –2 5) . Clinical protocol for cutaneous tumors and skin metastasis have been recently updated, as reported by the authors in ref 8. Of particular relevance is the need to progress in the treatments of deep seated tumors (Campana et al., European Journal of Surgical Oncology 45 (2019) 92-102).

The manuscript reports on combined effect of simultaneous administration by electropotration of bleomycin and a plasmid DNA (reporter gene) into CHO cell line.

The paper follows a previous one published by the same authors (cited in the References as 27), where enhancement of drug delivery by electrotransfer coupled to extracellular DNA was partially investigated.

The study has practical significance in vitro, providing experimental basis for further investigation in vivo. It is well designed and conducted, with results well supported in the discussion and an interesting conclusion.

Minor points to be clarified:

Neither in the figure 2, nor in the legend or in the manuscript there is any mention of experiment controls. Figure 2 shows the results on cells viabilty and DNA damage respect to Bleomycin concentrations used (Panel A and B). I suggest to add in the graphics the experimental control conditions: EP alone, and EP + plasmid vector.

Please revise Mat and Met and, where missed, provide city and country of manufacturers for chemicals and instruments used in the study (i.e.: Sigma Aldrich; Prep kit Qiagen; Spectrophotometer Nanodrop; Electroporator BTX T820; Scanner Canon; etc.).

Line 181 reports reference 37 and Line 198 reports reference 39: there is no reference 38 in between. Reference 38 is reported in Line 199.

Line 199-200: Please, add a reference at the end of the sentence: "However, the overstimulation of immune system by electroporation mediated immunotherapy could prove deleterious".

Author Response

REVIEW REPORT-1

 Comments and Suggestions for Authors

Improvements of electrochemotheray (ECT) protocols are of pivotal importance for clinical applications. In 2006 European Standard Operating Procedures of ECT (ESOPE) were released (Mir et al., E J C  S 4 ( 2 0 0 6 ) 1 4 –2 5) . Clinical protocol for cutaneous tumors and skin metastasis have been recently updated, as reported by the authors in ref 8. Of particular relevance is the need to progress in the treatments of deep-seated tumors (Campana et al., European Journal of Surgical Oncology 45 (2019) 92-102).

The manuscript reports on combined effect of simultaneous administration by electropotration of bleomycin and a plasmid DNA (reporter gene) into CHO cell line.

The paper follows a previous one published by the same authors (cited in the References as 27), where enhancement of drug delivery by electrotransfer coupled to extracellular DNA was partially investigated.

The study has practical significance in vitro, providing experimental basis for further investigation in vivo. It is well designed and conducted, with results well supported in the discussion and an interesting conclusion.

Minor points to be clarified:

Neither in the figure 2, nor in the legend or in the manuscript there is any mention of experiment controls. Figure 2 shows the results on cells viability and DNA damage respect to Bleomycin concentrations used (Panel A and B). I suggest to add in the graphics the experimental control conditions: EP alone, and EP + plasmid vector.

Thank You very much for the comment. The presented graph already has the requested one control i.e EP alone (represented by 0 nM bleomycin groups) – 0 nM point on the BLM+EP line is the same as EP alone as no bleomycin was added; likewise, 0 nM point on the BLM+pDNA+EP line is the same as EP+plasmid vector. We updated the figure and caption to outline this better.

Please revise Mat and Met and, where missed, provide city and country of manufacturers for chemicals and instruments used in the study (i.e.: Sigma Aldrich; Prep kit Qiagen; Spectrophotometer Nanodrop; Electroporator BTX T820; Scanner Canon; etc.).

The manuscript was updated according to the reviewers’ suggestions.

Line 181 reports reference 37 and Line 198 reports reference 39: there is no reference 38 in between. Reference 38 is reported in Line 199.

The references were rearranged to be in sequence

Line 199-200: Please, add a reference at the end of the sentence: "However, the overstimulation of immune system by electroporation mediated immunotherapy could prove deleterious".

Thank You for the comment. As we revised the manuscript to accommodate the changes requested by the other reviewers, we significantly rewrote and reorganized the part of the manuscript stating this. Therefore, the requested reference (Schulz-Knappe and Lautscham, 2018) could be found at line no. 62 of the revised manuscript.

Reviewer 2 Report

The submission has developed a novel method for controlled gene expression, which is interested, while there are many concerns need to be address before publication.

1: The authors should reorganize the writing to better display the novelty;

2: The quality of the figures should be enhanced significantly;

3: There are only 5 figures in this article. In my opinion, it is not enough to validate the novelty the authors claimed.  

Author Response

REVIEW REPORT-2

Comments and Suggestions for Authors

The submission has developed a novel method for controlled gene expression, which is interested, while there are many concerns need to be address before publication.

1: The authors should reorganize the writing to better display the novelty;

Thank You very much for the suggestion. We have included extra sections in the introduction and discussion parts of the manuscript to showcase the novelty of our suggested method better.

2: The quality of the figures should be enhanced significantly;

The resolution of the figures was in a good quality, however during the submission the resolution of the figures was reduced to save the size. According to the editor the submitted figures have sufficiently high resolution. The figures will be replaced with high quality pictures.

3: There are only 5 figures in this article. In my opinion, it is not enough to validate the novelty the authors claimed.  

Thank You for the comment. For this goal, we show a) The increased bleomycin cytotoxicity when the plasmid is present in the external medium during electroporation b) The increased DNA damage by bleomycin when the plasmid is present in the external medium during electroporation c) feasibility of achieving DNA electrotrasnfection with DNA-degrading bleomycin simultaneously present in the solution with the DNA and d) the control of DNA electrotrasnfection via bleomycin concentration. As this article is a proof-of-concept, the control we show can still be improved by optimizing the conditions of the treatment. Nevertheless, we believe that the data shown adequately represents bleomycin-controlled temporally resolved DNA electrotransfection. In response to the comment, we also improved parts of the results and discussion.

Reviewer 3 Report

This manuscript by Chopra et al. describes a strategy to combine gene electrotransfer and electrochemotherapy. In this way, it would be possible control the duration of electrotransfer-based gene expression by the cytotoxic effect of the chemotherapeutic drug bleomycin, which could provide a systemic effect in this approach due to immune activation. The work is interesting and is generally well presented. However, there are a few issues that should be addressed before publication.

Major comments:

1.       While the interaction of bleomycin with the plasmid employed is discussed in the context of drug uptake, it is unclear to me whether part of the effect seen on the DNA damage experiment (Figure 3) might actually be attributed to this introduced plasmidic DNA (either intact or damaged). From the image, it is unclear to me if pDNA was present in the control in the lower part of the Figure (which should have been present, in my opinion). I also believe there is a typographical error in the captions of all the images at the bottom part of the figure (should be X nM BLM + pDNA + EP, instead of X nM + BLM + pDNA + EP).

2.       Is there any plasmid that the authors believe could be particularly useful to test the therapeutic potential of this strategy in the future? If so, I believe this should be discussed in the manuscript, to also illustrate its future potential.

3.       Some discussion should be included about the ideal time frame for gene expression in the therapeutic context that the authors are proposing, that is, how long should the gene expression last to enable a therapeutic effect. Does it match with the period of gene expression obtained in this study? Should it be longer or shorter? How else could the results obtained here be improved to fit into these ideal conditions? (Including references to relevant work when necessary).

Minor comments:

4.       Is there any particular reason behind the choice of flow cytometry to obtain the absolute number of cells at different time points instead of more commonly used methods such as MTS or Alamar Blue assay?

5.       Due to the location of the experimental section being after the Results and Discussion, some concepts have not being properly introduced. For example, CA is used in the Results section, but it is only defined much later. The plasmid used is not specified before describing its transfection efficiency.

6.       In Figure 5, left panel, I am not sure if what they are showing could be described as transfection efficiency, since they are also measuring at different time points. Maybe “Percentage of GFP-Positive cells” or something similar could describe this better.

7.       The quality of the Figures is very low, and this aspect should be improved.

Author Response

REVIEW REPORT-3

Comments and Suggestions for Authors

This manuscript by Chopra et al. describes a strategy to combine gene electrotransfer and electrochemotherapy. In this way, it would be possible control the duration of electrotransfer-based gene expression by the cytotoxic effect of the chemotherapeutic drug bleomycin, which could provide a systemic effect in this approach due to immune activation. The work is interesting and is generally well presented. However, there are a few issues that should be addressed before publication.

Major comments:

  1. While the interaction of bleomycin with the plasmid employed is discussed in the context of drug uptake, it is unclear to me whether part of the effect seen on the DNA damage experiment (Figure 3) might actually be attributed to this introduced plasmidic DNA (either intact or damaged). From the image, it is unclear to me if pDNA was present in the control in the lower part of the Figure (which should have been present, in my opinion). I also believe there is a typographical error in the captions of all the images at the bottom part of the figure (should be X nM BLM + pDNA + EP, instead of X nM + BLM + pDNA + EP).

Thank You for the comment. Indeed, as reviewer notes, part of DNA damage can be attributed to the plasmid DNA partially digested by bleomycin. Nevertheless, we believe that this part is negligible. Firstly, no DNA damage evaluated by comet assay was seen after pDNA electrotransfer in the absence of bleomycin. Therefore, intact plasmid DNA is not seen in the tails of the comets. Secondly, we did the calculations that allowed us to estimate the amount of plasmid DNA delivered per cell and compare it to the amount of genomic DNA present in the CHO cells that were used in the experiments. For this, we started with calculating the volume from which the plasmids can interact with the cell. We used the formula presented in Pavlin & Miklavčič (2015) to determine the distance travelled by the plasmid during the electric pulses; it was equal to 2.73 × 10-7 m. As we had previously microscopically measured the cell size to be 9.7 × 10-6 m, the volume from which the plasmids are eligible to contact and interact with the cell plasma membrane is 8.09 × 10‑9 m3. For the pGFP plasmid (3486 bp) that we used, this equals 4523 plasmid molecules/cell. It should be noted that the amount of plasmids actually entering the cell will actually be much lower, because a) the possibility of plasmid DNA/cell membrane contact does not directly translate to DNA molecule entering the cell as the DNA molecules enter the cell after electroporation during a complex process involving membrane/DNA complex formation on the electroporated membrane and b) our calculations assume complete permeabilization of the plasma membrane, while in real conditions, only a part of plasma membrane reaches high enough induced transmembrane voltage to be electroporated. Therefore, the number presented here reflects unrealistically high number of plasmids. However, even if we consider this overestimation to be true, the total plasmid DNA length per cell would be 1.58 × 107 bp. Meanwhile, the genome length of Cricetulus griseus (Chinese Hamster) is 2.32 × 109 bp according to the representative genome assembly in NCBI database. Therefore, the length of genomic DNA is at least 147 times larger than that of the plasmid DNA, making plasmid DNA contribution to the comet tails insignificant. We also included these calculations into the manuscript for clarification (see lines 137-150 of the revised manuscript).

Regarding the second part of the comment, we have updated the figure to include the pDNA+EP control, and corrected the typographical errors in the figure.

  1. Is there any plasmid that the authors believe could be particularly useful to test the therapeutic potential of this strategy in the future? If so, I believe this should be discussed in the manuscript, to also illustrate its future potential.

As suggested, we included the discussion about potential therapeutic genes that could be introduced using our approach (Lines 266-268).

  1. Some discussion should be included about the ideal time frame for gene expression in the therapeutic context that the authors are proposing, that is, how long should the gene expression last to enable a therapeutic effect. Does it match with the period of gene expression obtained in this study? Should it be longer or shorter? How else could the results obtained here be improved to fit into these ideal conditions? (Including references to relevant work when necessary).

Thank You very much for a thought-provoking comment. Regrettably, the in vitro data presented here might not be enough to infer about the ideal timing for immune response as the system we used for this proof-of-concept study does not account for cell-to-cell interactions that are present in the more complex in vivo systems. To the extent of our knowledge, the investigations of such kind have not been conducted yet and pose an interesting question for future studies. However, caution must be applied as the immune response might be highly variable from species to species and from individual to individual, therefore, the transferability of the data gained from in vivo studies to clinical experiences might be complicated. These concerns notwithstanding, we have updated the discussion to reflect some of the considerations about the dynamics of gene expression and their implications for cancer therapy. (Lines 274-280).

Minor comments:

  1. Is there any particular reason behind the choice of flow cytometry to obtain the absolute number of cells at different time points instead of more commonly used methods such as MTS or Alamar Blue assay?

The effect on the treatment on the cells can be evaluated by many different methods such as proliferation assays (e. g., clonogenic assay), metabolic assays (e. g., MTT, MTS, Alamar Blue, XTT assays) or the methods evaluating absolute number of cells (e. g., flow cytometry or Coulter counter). Clonogenic assay is not suitable for the estimation of cell viability dynamics because the results show only the cell viability after 6-7 days (endpoint viability). The metabolic assays, like the MTS and Alamar Blue assays that the reviewer suggested, both reflect the cell metabolic activity rather than the absolute number of cells. Therefore, lower number of cells with higher metabolic activity could give the same results as higher number of cells with lower metabolic activity, leading to misinterpretation of the test results (For more on this, see Jakštys et al., 2015). Meanwhile, the flow cytometry assay employed in this study allowed the precise determination of the number of cells in the sample and was not affected by the metabolic state of the cells, leading to more precise results at specific time points post treatment.

  1. Due to the location of the experimental section being after the Results and Discussion, some concepts have not being properly introduced. For example, CA is used in the Results section, but it is only defined much later. The plasmid used is not specified before describing its transfection efficiency.

We have clarified the paper to introduce the abbreviations before they are used and included the plasmid details in the results section.

  1. In Figure 5, left panel, I am not sure if what they are showing could be described as transfection efficiency, since they are also measuring at different time points. Maybe “Percentage of GFP-Positive cells” or something similar could describe this better.

We have updated the graphs to better reflect the results shown.

  1. The quality of the Figures is very low, and this aspect should be improved.

The resolution of the figures was in a good quality, however during the submission the resolution of the figures was reduced to save the size. According to the editor the submitted figures have sufficiently high resolution. The figures will be replaced with high quality pictures.

Round 2

Reviewer 3 Report

The manuscript has been significantly improved, and it is now suitable for publication.